# The Effect of Corrective and Encouraging Accumulated Vibrotactile Feedback on Work Technique Training and Motivation—A Pilot Study

**DOI:** 10.3390/ijerph20186741

**Published:** 2023-09-11

**Authors:** Charlotta Langenskiöld, Annelie Berg, Liyun Yang

**Affiliations:** 1Unit of Occupational Medicine, Institute of Environmental Medicine, Karolinska Institutet, Solnavägen 4, SE-113 65 Stockholm, Sweden; 2Division of Ergonomics, School of Engineering Sciences in Chemistry, Biotechnology and Health, KTH Royal Institute of Technology, Hälsovägen 11C, SE-141 57 Huddinge, Sweden

**Keywords:** manual handling training, accumulated vibrotactile feedback, motivation, MSD prevention, physical ergonomics, smart workwear system

## Abstract

Encouraging feedback is shown to increase motivation and facilitate learning in different settings, though there is a lack of knowledge of applying it in work technique training. This pilot study aimed to evaluate two accumulated vibrotactile feedback strategies for work technique training using a smart workwear system. Eight women and two men participated in the study. They were divided into two groups, receiving the corrective feedback or the combined corrective and encouraging feedback while doing simulated manual handling tasks in a lab environment. Questionnaires and semi-structured interviews were used to evaluate the motivation, learning, and user experiences. In this small sample size, we saw that both groups significantly improved their work technique of upper arm and trunk postures, and no significant difference between groups was seen. In addition, both groups reported increased ergonomic awareness, were satisfied with the feedback training, and considered the system useful. However, the combined feedback group had slightly lower ratings of motivation and more negative experiences of the corrective feedback itself compared to the corrective feedback group. Both groups had positive experiences with the encouraging feedback. Future research should consider investigating the long-term learning effects of using solely corrective or encouraging accumulated feedback for work technique training with such systems.

## 1. Introduction

Musculoskeletal disorders (MSDs) are the most common work-related health problems within the European Union (EU). MSDs concern workers in all areas of work [1], and evidence shows a link between musculoskeletal injuries and manual handling tasks [2]. According to the EU Framework Directive on the manual handling of loads, the employer must ensure that workers receive adequate training and information about correct work techniques and ergonomic risks in the work environment whenever the manual handling of loads cannot be avoided [3]. Research within manual handling showed that training programs aimed at reducing the prevalence of work-related MSDs are ineffective in reducing the risk of injury and ill health in the long term [2,4,5,6]. One possible reason could be that the training programs did not lead to a lasting behavior change. Another possible reason could be the difficulty of transferring the training into the job [2,5]. Individual coaching focusing on the correct execution of exercises and motivation is described as a necessary factor for a successful training program [4].

Biofeedback can give workers information regarding ergonomic risks (external feedback) before any symptoms or discomforts in the body appear (internal feedback). In this way, workers can modify their work techniques and lower the risk of developing work-related MSDs before it becomes a health issue [7]. Biofeedback has been shown to be effective within ergonomic training in lifting technique [8], patient handling [9,10,11], and manual handling [12,13,14]. It has been shown to increase ergonomic awareness regarding the connection between work technique and potential risks [12,15]. Biofeedback training has also been shown to be effective in educating participants to find their own solutions regarding workstation design [13,14], which is an important part of participatory training [16,17]. In most of these studies, a concurrent corrective feedback strategy was used, which means the feedback is provided in real time when a movement above ergonomic thresholds occurs. Concurrent vibrotactile feedback is considered more suitable for shorter training sessions, as concurrent corrective feedback might lead to a dependency on external feedback, which complicates the learning process [13,14,18]. Another consideration was that ergonomic risks also have a cumulative character, and brief, non-neutral postures are not necessarily associated with increased risks. However, research regarding giving feedback on the accumulated load, which is referred to as accumulated feedback, is scarce within manual handling. How to use accumulated feedback and understand how it affects training and learning are of great interest because of its possibility to provide feedback at certain time intervals and hence provide a more sensible understanding of the accumulated ergonomic load. 

Another interesting aspect to consider when designing work technique training is how it affects the trainees’ motivational state. Motivation has been shown to be positively correlated with learning in different educational settings [19,20], in academic learning [21], within motor learning [22,23,24], and proven to have an essential influence on training transfer within a work environment [25,26,27]. According to the self-determination theory [20], motivation is a multidimensional concept that can be divided into autonomous or controlled motivation. Autonomous motivation is self-determined; it can be intrinsic, which is an action performed out of inherent interest and joy, or extrinsic, which is an action performed out of perceived value [20,25]. On the other hand, controlled motivation is described as an action performed due to external rewards, pressure, or punishments [20,28]. Both autonomous and controlled motivations are positively associated with behavior change [26,27], whereas controlled motivation may lack a lasting effect on behavioral outcomes [25,27,28].

Two aspects of training have been shown to promote autonomous motivation and facilitate learning transfer: the experience of competence [28] and the perceived usefulness [16,25,27]. Positive or encouraging feedback is shown to enhance the experience of competence, which further facilitates intrinsic autonomous motivation, i.e., inherent interest and enjoyment [20,29,30,31]. Negative or corrective feedback has been argued to undermine intrinsic motivation as it is considered ego-threatening and hence fails to facilitate feelings of competence [29]. Still, if corrective feedback includes information on how to improve and supports an individual in reaching their goals, it can enhance feelings of competence and consequently facilitate intrinsic motivation [29]. For uninteresting tasks, especially when there is a lack of inherent joy and interest, it is important to provide individuals with a meaningful rationale and understand the value of the activity, which facilitates autonomous extrinsic motivation [32]. Perceived usefulness is shown to promote extrinsic autonomous motivation and facilitate learning transfer [16,25,27] and has also been described as a better predictor of persistence for uninteresting activities [33]. Therefore, the motivational aspects should be considered when designing work technique training with feedback for better learning. However, there is a lack of knowledge in the use of corrective- and encouraging feedback in work technique training and its effects on motivation for learning. At the same time, the amount of feedback that workers receive during a training program can be an important factor to lead to behavior change. Giving solely encouraging feedback may not ensure adequate exposure to the feedback during work technique training, depending on the tasks and workers’ previous knowledge of ergonomics. Considering the explorative nature of the study, a combination of corrective and encouraging feedback was of interest to be tested in comparison to solely corrective feedback as one step towards using encouraging feedback in work technique training.

Therefore, this study aimed to investigate and compare the effects of two different accumulated vibrotactile feedback strategies, i.e., corrective feedback and a combination of corrective and encouraging feedback, on work technique training for the dominant upper arm and trunk. More specifically, the ergonomic behavior, the levels of motivation, as well as the user experiences were examined and compared between the two groups.

## 2. Materials and Methods

### 2.1. Participants

Ten participants (8 women and 2 men) took part in the study during March/April 2021, with a mean age of 43.9 (SD 12.0), height of 166.6 cm (SD 9.4), and weight of 74.2 kg (SD 10.6). Exclusion criteria were restrictions in movement or pain from the dominant shoulder/arm, back, hip, or knees. Among them, nine participants were office workers whose work consisted mainly of administrative tasks, and one participant was an industrial worker whose work consisted mainly of manual handling. The participants were divided into two groups, with a strategy that participants of the same gender were separately allocated into two groups one by one to ensure an even gender distribution between the two groups. Informed consent was obtained from each subject prior to the study. The study was approved by the Regional Ethics Committee in Stockholm (Dnr, 2019-01206).

### 2.2. Study Settings

Prior to the experiment, the participants received verbal instructions, all from the same trial leader, regarding basic physical ergonomics. Instructions included try (i) working with arms close to the body and (ii) minimizing forward bending and twisting of the back. The participants also received verbal information regarding the accumulated vibrotactile feedback that they were about to receive during the task. They were informed that it was given at a one-minute interval and based upon the work technique adopted during the last minute of work. 

The participants were able to perform one practice session before the start of the experiment. The practice consisted of performing parts of the task—this was carried out with no vibrotactile feedback. During the practice session, participants who obviously did not seem to adopt the previous verbal instructions were reminded of the instructions again. The experiment included three sessions in a controlled lab environment: a baseline, an intervention, and a post-intervention (Figure 1). The baseline consisted of one work cycle (4–6 min) with no feedback. The intervention consisted of two work cycles with accumulated vibrotactile feedback, where group A was provided with the corrective feedback strategy and group B with the combined feedback strategy. The post-intervention consisted of one work cycle with no feedback. 

Within each work cycle, participants performed four types of manual handling tasks: sorting 38 documents, lifting 8 large empty boxes, moving 8 ring binders, and lifting 10 small empty boxes (Figure 2). The tasks were designed as situations where upper arm lifting and trunk bending would usually occur, and the participants had the possibility to receive feedback during the training. The participants performed the tasks in a predetermined order following a list with instructions, and they were informed to use their dominant hand if the task could be performed with one hand.

### 2.3. The Feedback and Measurement System

The smart workwear system’s vibrotactile feedback module was used to provide the vibrotactile feedback and measure postural data of the upper arm and trunk [14]. It consists of a customized workwear t-shirt with embedded pockets, two Inertial Measurement Units (LPMS-B2 IMU, LP Research, Tokyo, Japan), two customized vibration units, and an Android smartphone application (Figure 3). The IMUs, which are placed in the pockets on the upper arm and upper back, communicate with the Android smartphone via Bluetooth in real time (Figure 4). The Android application calculates the trunk flexion/extension angle and upper arm elevation angle relative to the reference position (0°) using the quaternion signals, which incorporate accelerometer and gyroscope data using the sensor’s built-in fusion algorithm. The vibrotactile units, which were placed in the pockets on the upper arm and chest, received signals from the Android application via Bluetooth and provided vibrotactile feedback according to the pre-set ergonomic thresholds and feedback strategy (see more description in Section 2.3.1 and Section 2.3.2). 

Before starting a measurement, a calibration procedure was performed with each participant to set the reference position with the system. For the trunk, the participants were asked to stand straight in a relaxed position with the arms hanging down and hold still when the trunk calibration posture was set in the Android application. For the upper arm, the participant was instructed to sit down with the arm hanging over the back of a chair and hold a 2 kg dumbbell for two seconds [34] when the arm calibration posture was set. 

#### 2.3.1. Feedback Strategy

Two types of feedback strategies were included in the system: (i) accumulated corrective vibrotactile feedback (referred to as “corrective feedback” in the following text) and (ii) accumulated corrective and encouraging vibrotactile feedback (referred to as “combined feedback” in the following text). Two types of vibration were provided by the vibration unit: the corrective feedback was given as a persistent vibration lasting two seconds each time it is triggered, designed to mimic a wasp; the encouraging feedback was given as two repetitions of three short vibrations each being on and off for 0.1 s, and a 0.5-s pause between repetitions, designed to mimic bird songs. The vibration choice was discussed and tested by the research team to make sure that the two types of feedback were easily distinguishable. Both types of vibrations were given with the full intensity by the vibration unit. Since the experienced vibration was affected by the placement of the whole vibration unit in the pockets of each participant, which was not measured, the information on vibration intensity was not available in this study. For the “corrective feedback” strategy, only corrective vibrotactile feedback was provided, and no feedback was provided if the participants performed within the threshold. For the “combined” strategy, corrective vibrotactile feedback was provided when the pre-set threshold was exceeded, and encouraging vibrotactile feedback was provided if the participants performed within the thresholds.

#### 2.3.2. Ergonomic Thresholds

Accumulated feedback was used and given at every minute in this study. For the dominant upper arm, the feedback threshold was set at exceeding 30° elevation for more than 30% of the time (Figure 5), which was based and modified on the recommended action levels by Arvidsson and colleagues [35]. The ergonomic threshold was decided to be lowered from the recommended action levels, based on previous pilot testing, to ensure that participants would receive feedback during the training. For the trunk, the feedback threshold was set at exceeding 30° forward bending for more than 10% of the time (Figure 5), based on findings from Lötters and colleagues [36]

### 2.4. Questionnaire on Motivation

A questionnaire on motivation consisting of three themes, i.e., perceived competence, interest and joy, and value and usefulness, was used to study the motivation after the training in this study. The questionnaire was based on the Intrinsic Motivation Inventory (IMI) [37], translated into Swedish based on a validated Swedish version of the questionnaire [38], and modified to work technique training. The IMI is considered flexible, i.e., relevant themes can be chosen depending on the research need without affecting the reliability of the measurement device [37]. Each theme consisted of three statements and was scored on a 7-point Likert scale from 1, “strongly disagree”, to 7, “strongly agree”. Statements included the following: (i) Theme—Interest and enjoyment: 1. “I thought this activity was quite enjoyable”, 4. “I thought this activity was a boring activity (R)”, and 7. “I would describe this activity as very interesting”; (ii) Theme—Perceived competence: 2. “I think I am pretty good at this activity”, 5. “I am satisfied with my performance at this task”, and 8. “After working at this activity for a while, I felt pretty competent”; and (iii) Theme—Value and usefulness: 3. “I believe this activity could be of some value to me”, 6. “I think that doing this activity is useful for work technique training”, and 9. “I think doing this activity could help me to work in a safer way”. 

### 2.5. User Experience Interview

A semi-structured interview was conducted to assess the participants’ user experience of the wearable system and feedback training (see the interview guide in Appendix A). The questions focused on whether the participants experienced discomfort (i.e., the internal feedback) and its impact on their work technique; participants’ experience of the corrective and encouraging feedback; whether the participants’ work technique was changed due to the vibrotactile feedback (i.e., the external feedback); potential learnings from the feedback; willingness to use the system during a working day; and any other comments. 

### 2.6. Data Analysis

Shapiro–Wilk test was used to check the normality of the data. A *p*-value < 0.05 was chosen for statistical significance using IBM SPSS Statistics 27 (Armonk, NY, USA).

#### 2.6.1. Work Technique Data

To analyze the effect of the accumulated vibrotactile feedback on the training of work technique, the intervention and post-intervention sessions were compared to the baseline for the whole group (within-subjects comparison). The following parameters, which have been associated with MSDs and used in previous studies, were included for the arm elevation: percentage of time ≥30°, ≥45°, and ≥60° and the 50th, 90th, and 99th percentile of arm elevation [14,35,39,40]; for the trunk inclination: percentage of time ≥20°, ≥30° and ≥45° and the 50th, 90th and 99th percentile of the trunk inclination [14,36,41,42]. Paired *t*-tests were used for normally distributed data, and Wilcoxon signed-rank tests were used for non-normally distributed data. To analyze the difference in the training effects between the two feedback strategies, independent *t*-tests were used to compare the changes in the intervention and post-intervention sessions compared to the baseline between the two groups (between-subjects comparison). The same parameters were used as listed above.

#### 2.6.2. Motivation Questionnaire

The nine items regarding motivation were graded on a Likert scale from 1 = strongly disagree to 7 = strongly agree. The participants’ average scores were calculated for every theme from the three statements. The negatively worded statement (item 4) was reversed and calculated so it matched with the scoring. As the data were normally distributed, independent *t*-tests were used for comparison between the two groups. As the questionnaire had been modified from the validated Swedish version, a Cronbach’s alpha test was carried out to investigate its reliability. A Cronbach’s alpha value between 0.7 and 0.95 is considered satisfactory [43,44].

#### 2.6.3. User Experience

The interviews were recorded, transcribed, and analyzed through thematic analysis [45] and set into different themes. The results were also presented in a quantified way when appropriate.

## 3. Results

### 3.1. Effects on Work Technique

#### 3.1.1. The Changes in Arm Elevation

As shown in Table 1 and Figure 6B, the whole group spent significantly less time in upper arm elevations ≥ 45° and ≥60° during intervention and post-intervention compared to baseline when training with accumulated vibrotactile feedback. It also shows that significantly less time was spent with the arm elevated ≥30° during post-intervention compared to baseline. A significant decrease was also seen in the 50th and 90th percentile of arm elevation (Table 1 and Figure 6A). No significant decrease in arm elevation was seen at ≥30° during intervention compared to baseline, nor in the 99th percentile of arm elevation.

There was no significant difference in the change of upper arm angles between the group receiving corrective feedback and the group receiving combined feedback during the intervention or post-intervention compared to baseline, with all *p*-values > 0.05 (Table 2). A small trend could be observed that the group with combined feedback had a larger reduction of arm elevation in most of the parameters, both during intervention and post-intervention, compared to baseline (Table 2). For example, the time in upper arm elevations ≥ 30° had an average of 9.2% reduction during post-intervention compared to baseline for the group combined feedback, whilst the group corrective feedback had an average of 4.7% reduction.

#### 3.1.2. The Changes in Trunk Inclination

As shown in Table 3 and Figure 6D, a significant decrease in the percentage of time was spent with the trunk in forward inclination angles ≥ 30° and ≥45° during intervention compared to baseline. During post-intervention, significantly less time was spent with the trunk in forward inclination angles ≥ 20°, ≥30° and ≥45° compared to baseline. A significant decrease in time spent with the trunk in forward inclination was also seen in the 90th and 99th percentile (Table 3 and Figure 6C).

There was no significant difference in the change of trunk inclination between the group receiving corrective feedback and the group receiving combined feedback during the intervention or post-intervention compared to baseline, with all *p*-values > 0.05 (Table 2). No distinctive trend was observed when comparing the reduction of trunk inclination between the groups.

### 3.2. Measurement of Motivation

The internal consistency was calculated using Cronbach’s (alpha) statistic. The results showed acceptable reliability for all themes, interest and joy (0.809), perceived competence (0.822), value and usefulness (0.783), and total motivation, where all themes were included (0.904). No significant difference in the level of motivation between the two groups was seen in each theme. The mean (SD) values of the scores for the two groups were (i) interest and joy (group corrective feedback, 5.6 (1.1); group combined feedback, 4.8 (1.9); *p* = 0.44), (ii) perceived competence (group corrective feedback, 5.3 (0.9); group combined feedback, 4.5 (1.1); *p* = 0.29), and (iii) value and usefulness (group corrective feedback, 6.1 (0.9); group combined feedback, 5.5 (1.2); *p* = 0.4). Under the theme “value and usefulness”, all participants replied that the training could be useful for work technique training, and the majority (n = 9) believed the training could help them work in a safer manner.

For the overall motivation, the corrective feedback group showed a slightly higher score with a mean (SD) of 5.7 (0.8) compared to the combined feedback group, 5.0 (1.3). The mean value of difference was 0.7 on a scale of 1 to 7 between the two groups. However, the difference was not statistically significant, *p* = 0.33.

### 3.3. Semi-Structured Interview

**Ergonomics awareness:** A majority of the participants (n = 9) expressed an increase in ergonomics awareness. The biofeedback training made the participants think about their everyday work tasks and the work techniques they adapted to perform them. The training aroused thoughts about shortcomings in their own work technique as well as thoughts about physical ergonomics playing a more important role than one would think.


*“That you should try to think about ergonomics in everyday life, at work. That it probably means more than you think.”*


**Experience of the feedback:** The experiences of the two different feedback strategies were different between the groups. The participants that only received corrective accumulated feedback described the feedback as positive (n = 2) or neutral (n = 3), using words like “good” and “unaffected”. 

The participants who received combined feedback experienced the corrective feedback more negatively compared to the group that only received corrective feedback. The combined feedback group (n = 5) used words like “stressing” and “irritating” when describing the corrective feedback. The encouraging feedback was experienced in a positive manner (n = 4) using words such as “happy” and “motivated” when describing the feedback. Only one participant described the encouraging feedback as “distracting”.

**The system usability:** All participants (n = 10) were positive regarding using the system as a part of a working day and felt that they had learned something (n = 9) from the training.

**Interpretation difficulties:** The accumulated feedback made it complicated to understand what specific parts of the task had been performed in a right or wrong manner. The accumulated feedback was described as difficult to understand what precise physical load had caused the corrective or encouraging feedback (n = 7). It could also be a challenge to process the information communicated by the feedback system and simultaneously continue the work task.


*“Thought what I did wrong but didn’t understand what it was. Then I thought: what did I do wrong now? You’d like to know actually.”*


**Discomfort during training:** None of the participants experienced any musculoskeletal discomfort during the training. 

## 4. Discussion

The current study evaluated the effects of two accumulated vibrotactile feedback strategies, i.e., the corrective feedback and the combined corrective and encouraging feedback, on work technique training in simulated manual handling work tasks. The major findings of the present pilot study were that both groups (i) significantly decreased the accumulated time spent in awkward upper-arm and trunk positions in simulated manual handling tasks, (ii) increased the participants’ ergonomic awareness, and (iii) were considered of high usability for work technique training. In addition, (iv) the group receiving corrective accumulated vibrotactile feedback had slightly higher levels of motivation score (not statistically significant) and more positive experience of the vibrotactile feedback compared to the combined feedback strategy.

### 4.1. Changes in Work Technique

The reduction was seen in both upper arm elevations and trunk inclination during intervention compared to baseline and during post-intervention compared to baseline (Table 1 and Table 3). This indicates that there was an ongoing improvement in work technique when feedback was provided, and the improvement was maintained even after the feedback was withdrawn. Similar positive results have been seen in previous studies on the effects of concurrent vibrotactile feedback training within manual handling [12,13,14]. However, it is difficult to compare the extent of improvements from the current study with previous studies as both the performed work tasks and workstation designs, as well as the type of feedback, differed, which could largely affect the potential room for improvement. 

When comparing changes in work postures between the two feedback conditions, i.e., the corrective vs. combined accumulated vibrotactile feedback, no significant difference was seen in the intervention and post-intervention sessions (Table 2). These results indicated that participants in both groups were able to improve their work technique regardless of the feedback strategies provided.

### 4.2. Ergonomic Awareness

Both feedback training strategies increased the ergonomic awareness in the majority of participants (n = 9). Results from the semi-structured interview suggested that the training had triggered thoughts among participants about potential risks in their own work technique and concluded that ergonomic behavior was more important than previously thought. Biofeedback training can provide information about an individual’s ergonomic behavior even when internal feedback, i.e., discomfort or pain from the body, is not available. This is an important aspect of training for safe and healthy work postures before discomfort or pain occurs. As seen in this study, the biofeedback system provided information about potentially hazardous ergonomic behavior, which consequently supported participants in improving their work technique and led to a change in the perceived value of the behavior. The change in the value of the behavior is argued to enhance autonomous motivation, which was shown to be correlated to training transfer and an important supporting factor when designing work safety training [46].

### 4.3. The Accumulated Feedback 

Even though all participants significantly reduced the time spent in awkward upper arm and back postures, and the majority showed an increased level of ergonomic awareness, the accumulated vibrotactile feedback used in this study was described as difficult to understand (n = 7). Consequently, participants did not understand which specific work-related task had triggered the feedback. This contrasted with previous studies on concurrent vibrotactile feedback within manual handling, where the participants were able to understand the relationship between movement and received feedback and, hence, were able to identify, analyze, and solve ergonomic problems [13,14].

Still, the accumulated feedback is considered relevant for its purpose, considering several aspects, which are discussed here in the following text. The accumulated feedback could provide information regarding the potential risk of accumulated time spent in awkward postures and movements. It is worth mentioning that brief, non-neutral postures do not necessarily mean an increased risk of work-related musculoskeletal disorders. In addition, under certain conditions, it might be beneficial to have non-neutral postures as part of the variation in the work postures, e.g., in stretching movements or to avoid static postures. The accumulated feedback might also be more suitable during longer training sessions as the feedback frequency is lower. More importantly, it might decrease feedback dependency after the feedback withdrawal to improve long-term learning. Still, since the accumulated feedback strategy may increase the complexity of the training, as suggested by the results of this pilot study, a concurrent feedback strategy might be needed at the beginning of the training. The concurrent feedback might also be more suitable if the goal was to train workers to identify ergonomic risks and potentially solve ergonomic problems in their work. Therefore, the choice of feedback strategy should suit the training scenario and purpose.

### 4.4. Motivation

There was no statistically significant difference in the survey results on motivation between the two groups. The group receiving only corrective feedback tended to report higher levels of motivation compared to the group receiving both corrective and encouraging feedback. It is worth noticing that the interview results could show that participants in the corrective feedback group described the “corrective feedback” with positive or neutral words, whilst participants training with a combined feedback strategy described the “corrective feedback” in a more negative manner. However, in previous studies, the concurrent “corrective feedback” on work technique training within manual handling was described as positive [13,14]. The positive experience was similar to that of the group receiving corrective feedback in this study but contrary to the group with combined feedback. 

Research within education [19,20,21] and motor learning [31] showed that practice conditions with encouraging feedback increased the experience of competence and, hence, higher levels of intrinsic motivation that facilitated learning. Biofeedback has been described as a motivating source of feedback as it is considered an objective source of feedback, whilst verbal feedback is of a subjective nature and often fails to contribute to perceived competence and intrinsic motivation as it can be difficult to receive without hurting one’s pride [21]. Feedback based on objective information can be easier to accept compared to subjective feedback, which can trigger negative emotions, leading to a lower experience of competence and intrinsic motivation [21]. It is possible that the combination of both encouraging and corrective feedback affected the perception of the corrective feedback for the combined feedback group. Therefore, the objectivity of the feedback might have decreased when the encouraging feedback was combined with the corrective feedback. The combination added a positive (encouraging) and negative (corrective) aspect to the feedback strategy that can arguably be considered less objective than providing only corrective feedback.

One strength of this study was the evaluation of autonomous extrinsic motivation, especially in activities that are not seen as interesting but important, which is the common case in work technique training. This is in agreement with research on work motivation, showing that autonomous extrinsic motivation may lead to the most effective performance while performing uninteresting tasks that require discipline and engagement [33]. Work safety and ergonomic training might not be considered of high “interest and joy”, whereas ergonomic training with vibrotactile feedback can be considered of high “value and usefulness”, as indicated by both groups’ relative scores in the motivation questionnaire. They also stated that ergonomic behavior was considered of more importance after the training in the interview. This perceived value and usefulness of the activity could be positively correlated with training transfer [16], but this is not examined in the current study.

### 4.5. User Experience

All participants were satisfied with the training and felt they could use the smart workwear system during a part of a workday. Interestingly, Hogan et al. found in their systematic review on training transfer following manual handling training that 75% of participants were dissatisfied or negative towards the training [5]. Training satisfaction has been shown to be important for training transfer [25] and is hence of importance and should be taken into consideration when designing a training program within manual handling. Therefore, our results suggested that the training using the smart workwear system has advantages in obtaining a positive user experience.

### 4.6. Limitations and Future Research

There were several limitations that need to be considered when interpreting the results of this pilot study. A convenience sampling strategy was used in order to achieve data collection during spring 2021 under the pandemic. The gender distribution and career background of the participants were not equal, and the sample size was small, especially when comparing the two groups. We have examined the data regarding different genders and work experience in manual handling, and no deviation could be observed for the minor groups in the work technique data or the motivation level. The relatively small sample size could limit the statistical test results in finding significant differences. It is possible to see significant differences between the two groups if the sample size was larger, especially in the motivation level. Still, the results from the interviews provided richer information regarding the positive and negative user experiences. Therefore, the findings of this study could be of great value for designing future studies on work technique training with biofeedback, with possibly larger groups. 

In addition, the study was conducted in a lab environment over a short period, and there was no long-term follow-up of the training effects. Therefore, the effects in a real work environment and the learning transfer need to be examined in future studies. We consider this short-term lab study as a necessary first step to understanding the effects of accumulated feedback before larger studies look at its long-term effects in a real work environment.

Lastly, the study had no control group who did not receive any feedback. The consideration with a control group can be to eliminate the effects of learning by only repeating the task, e.g., learning from the body’s internal feedback, such as a feeling of discomfort. Since no participant reported feeling discomfort when performing the task, the internal feedback was perhaps not enough to lead to changes in the work techniques. In a previous study, repeating a manual handling task without feedback for three work cycles showed no improvement in the work techniques [14]. Therefore, considering the limited resources, the choice of two groups receiving the two types of accumulated vibrotactile feedback was prioritized in the study design.

Based on the findings of this pilot study, future research should consider evaluating the use of corrective and encouraging vibrotactile feedback separately. A design of the work tasks would need to be considered to ensure that both groups received enough exposure to the corrective or encouraging vibrotactile feedback. Studies on long-term learning with accumulated vibrotactile feedback are needed to better understand its effects on work technique training, motivation, and learning transfer. It could also be of interest to explore the effects of using a combination of concurrent and accumulated feedback as a work technique training strategy. For example, by starting with concurrent feedback (lower complexity) and then moving towards accumulated feedback (higher complexity), it could possibly enhance understanding of the feedback, e.g., which specific tasks and movements lead to the provided feedback, and the accumulated ergonomic risks involved over a period of work time.

## 5. Conclusions

This pilot study evaluated the effects of two feedback training strategies on work technique, individual motivation, and user experience. In this small sample size, we saw that training with both accumulated vibrotactile feedback strategies significantly improved the work technique of the upper arm and trunk. Participants also reported increased individual ergonomic awareness after training from both groups. The corrective feedback group showed slightly higher levels of motivation compared to the combined feedback group, but it was not statistically significant. The interview results suggested that the corrective feedback group had a more positive experience of the “corrective feedback” compared to the combined feedback group. In addition, both groups were satisfied with the feedback training and considered the system useful. Future research can look into the long-term effects of learning using corrective or encouraging accumulated feedback separately using such feedback systems with larger sample sizes.

## Figures and Tables

**Figure 1 ijerph-20-06741-f001:**
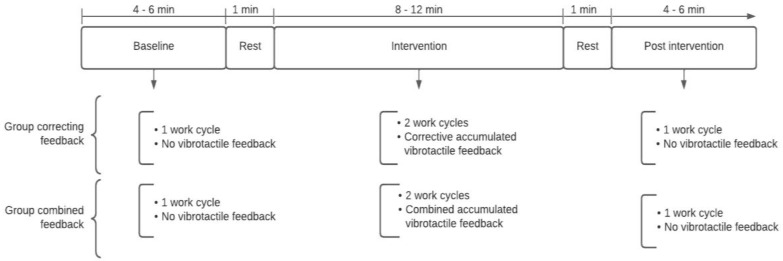
Flowchart of the study setting for group “corrective feedback” and group “combined feedback”.

**Figure 2 ijerph-20-06741-f002:**
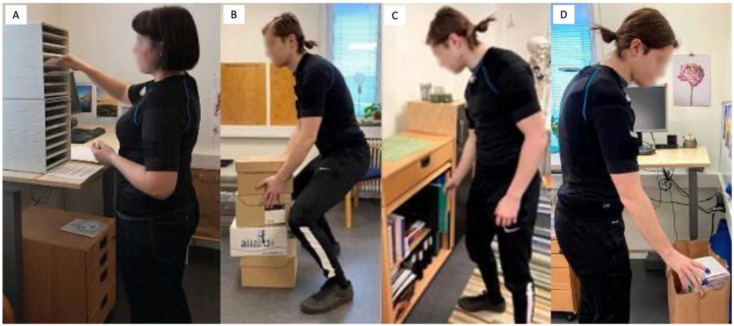
Simulated work tasks: (**A**) sorting documents, (**B**) lifting large boxes, (**C**) sorting binders, and (**D**) handling small boxes.

**Figure 3 ijerph-20-06741-f003:**
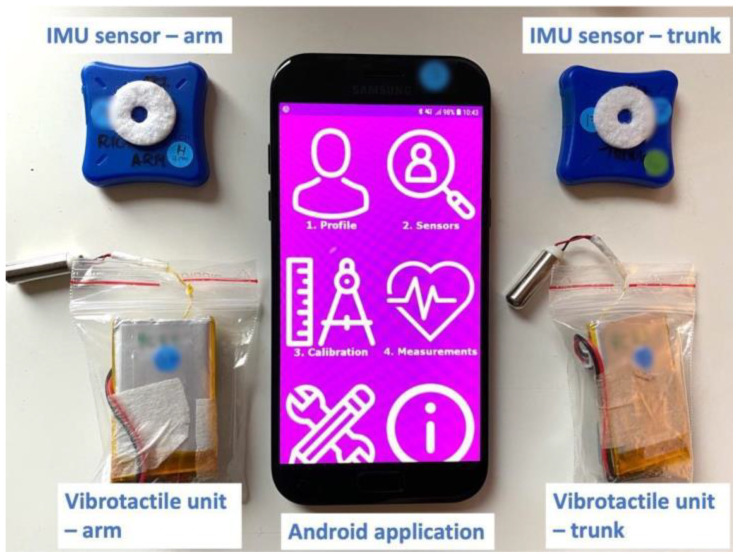
The smart workwear system vibrotactile feedback module composition: two IMU sensors and two vibrotactile units, one for the arm and one for the trunk accordingly, and one Android smartphone application.

**Figure 4 ijerph-20-06741-f004:**
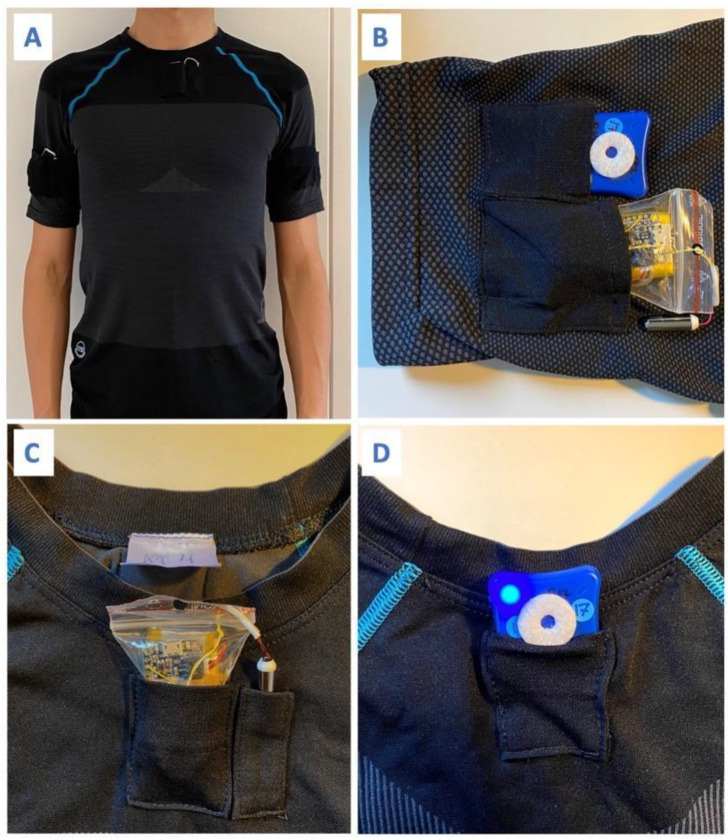
(**A**) The customized workwear t-shirt with embedded pockets. (**B**) The placement of the upper arm IMU and vibration unit. (**C**) The placement of the trunk vibration unit on the chest. (**D**) The placement of the trunk IMU on the upper back.

**Figure 5 ijerph-20-06741-f005:**
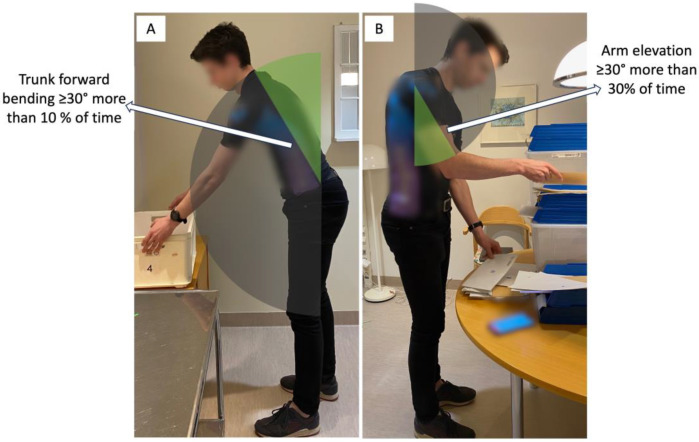
Illustration of the ergonomic thresholds used for the accumulated feedback. (**A**) For the trunk: exceeding 30° forward bending for more than 10% of the time. (**B**) For the arm: exceeding 30° elevation for more than 30% of the time.

**Figure 6 ijerph-20-06741-f006:**
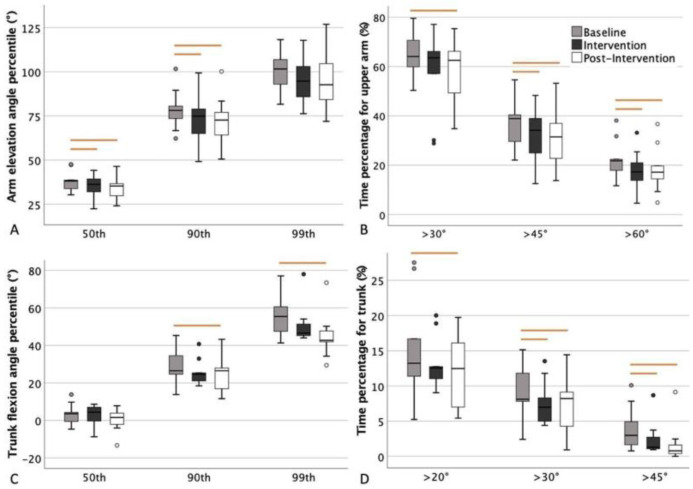
Arm elevation and trunk inclination of the whole group from baseline, intervention, and post-intervention sessions: (**A**) The 50th, 90th and 90th percentile arm elevation angles during each session. (**B**) The time percentage of upper arm elevation angle exceeding 30°, 45°, and 60°. (**C**) The 50th, 90th and 90th percentile trunk flexion angles. (**D**) The time percentage of trunk flexion angle exceeding 20°, 30°, and 45°. The boxplots show the median, the interquartile range, the max and min values, and the outliers. Significant differences between the sessions are marked with lines above the box plots.

**Table 1 ijerph-20-06741-t001:** A within-subjects comparison of arm elevations ≥30°, ≥45°, and ≥60° as well as the 50th, 90th, and 99th percentile during baseline, intervention, and post-intervention. Significant differences compared to baseline (*p* < 0.05) are presented in bold by the paired *t*-tests. The comparison includes 9 recordings due to incomplete data from one participant in the group combined feedback.

	Baseline	Intervention				Post-Intervention			
	Mean (SD)	Mean (SD)	Diff (%)	*p*-Value	95% CI	Mean (SD)	Diff (%)	*p*-Value	95% CI
Arm elevation (time %)									
≥30°	64.3 (9)	57.2 (16.7)	−11	0.064	−0.51–14.7	57.6 (14.7)	−10.4	**0.019**	1.4–12
≥45°	38 (10.8)	31.3 (13)	−17.6	**0.008**	2.3–11	32.4 (13.6)	−14.7	**0.002**	2.8–8.5
≥60°	22.1 (8.2)	17.8 (8.5)	−19.5	**0.002**	2.1–6.5	18.4 (9.6)	−16.7	**0.001**	2–5.5
Arm elevation (°)									
50th	37.9 (6.2)	34.6 (7.5)	−8.7	**0.016**	0.8–5.9	34.9 (7.8)	−7.9	**0.002**	1.5–4.5
90th	78.8 (11.7)	72.5 (14.8)	−8	**0.003**	2.8–9.6	72.2 (14.4)	−8.3	**<0.001**	4–9.2
99th	100.1 (11.1)	96.8 (14.2)	−3.2	0.099	−0.78–7.4	95.1 (16.8)	−5	0.062	−0.31–10.4

**Table 2 ijerph-20-06741-t002:** Changes in work technique of upper arm elevation and trunk inclination compared between the group corrective feedback and the group combined feedback. The changes were presented in the intervention session compared to baseline (to the left) and the post-intervention session compared to baseline (to the right). One arm angle data recording (in group combined feedback) and one trunk angle data recording (in group corrective feedback) were missing. No statistically significant difference was observed.

	Intervention–Baseline				Post-Intervention–Baseline			
Change in Value	Group Corrective Feedback	Group Combined Feedback			Group Corrective Feedback	Group Combined Feedback		
	Mean	Mean	*p*-Value	CI 95%	Mean	Mean	*p*-Value	CI 95%
Arm elevation (time %)								
≥30°	−2.6	−12.7	0.14	−24–4	−4.7	−9.2	0.37	−15.4–6.5
≥45°	−5.1	−8.6	0.36	−12.3–5.12	−4.2	−7.3	0.27	−9.4–3.2
≥60°	−3.3	−5.6	0.28	−6.6–2.1	−3.1	−4.5	0.4	−5.1–2.3
Arm elevation (°)								
50th	−2.1	−5	0.2	−7.7–1.9	−2.4	−3.7	0.4	−4.7–2.2
90th	−4.2	−8.9	0.15	−11.7–2.2	−5.6	−7.8	0.43	−8.8–4.5
99th	−2.4	−4.4	0.58	−10.5–6.5	−5.2	−4.8	0.94	−13.4–14.2
Trunk inclination (time %)								
≥20°	−2.2	−2.3	0.97	−2.7–−7.9	−1.4	−5.4	0.13	−9.4–1.5
≥30°	−2.2	−1.4	0.54	−2.3–4	−1.6	−3.2	0.34	−5.3–2.1
≥45°	−1.8	−1.2	0.59	−1.9–3.1	−2	−2.2	0.93	−3.9–3.6
Trunk inclination (°)								
50th	−2.5	−0.5	0.46	−4.1–8.1	−4	−2.6	0.69	−6.6–9.3
90th	−5	−2.5	0.48	−5.5–10.6	−4.6	−5.6	0.79	−9.6–7.6
99th	−8.7	−2.2	0.15	−3.0–16	−9.9	−11.1	0.8	−12.2–9.8

**Table 3 ijerph-20-06741-t003:** A within-subjects comparison of trunk inclination angles ≥ 20°, ≥30°, and ≥45° as well as the 50th, 90th, and 99th percentile during baseline, intervention, and post-intervention. *p*-values from the paired *t*-tests or the Wilcoxon signed-rank tests are presented comparing the intervention or post-intervention compared to baseline. The comparison included 9 recordings due to incomplete data from one participant in the group’s corrective feedback.

	Baseline	Intervention			Post-Intervention		
	Mean (SD)	Mean (SD)	Diff (%)	*p*-Value	Mean (SD)	Diff (%)	*p*-Value
Truck inclination (time%)							
≥20°	15.5 (7.4)	13.2 (3.8)	−14.8	0.11 ^w^	11.9 (5.1)	−23.2	**0.028 ^w^**
≥30°	9.5 (4.2)	7.7 (3.1)	−18.9	**0.026**	6.9 (4.3)	−27.4	**0.014**
≥45°	3.9 (3.2)	2.5 (2.5)	−35.9	**0.008 ^w^**	1.8 (2.9)	−53.8	**0.008 ^w^**
Truck inclination (°)							
50th	3.4 (5.8)	2 (6)	−41.2	0.32	0.2 (6.2)	−94.1	0.057
90th	29.4 (9.9)	25.8 (6.9)	−12.2	0.07	24.2 (9.7)	−17.7	**0.012**
99th	55.9 (10.8)	50.8 (10.7)	−9.1	0.066 ^w^	45.3 (12.3)	−19.0	**0.008 ^w^**

Significant differences (*p* < 0.05) are displayed in bold and Wilcoxon signed-rank test results are marked with ^w^.

## Data Availability

Data presented in the paper are available on request from the corresponding author, L.Y.

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
