# Peer review of "The Effect of Corrective and Encouraging Accumulated Vibrotactile Feedback on Work Technique Training and Motivation—A Pilot Study"

_ijerph, 2023, doi:10.3390/ijerph20186741_

Round 1

Reviewer 1 Report

This paper investigates the impact of corrective and encouraging vibrotactile feedback on monitoring and enhancing working posture. The research content of the paper is clear, and the research also has positive implications for the development of a posture feedback system. However, the research should be further explored based on the objectives of the study.

1. Why did the authors use corrective and encouraging vibrotactile feedback condition, not encouraging condition only? The reviewer concerns that there would be confounding effects of those two feedbacks, as it is described in the discussion.

2. Although not significant, the results of comparing feedback performance must be explained from the start of the Results section, not the Appendix; as readers expect the experimental output based on the title and objective of the study. 

3. The first paragraph of the Introduction includes lengthy information regarding MSDs, while the core issue of this study is vibrotactile feedback. The authors might reduce (or remove) the first paragraph for MSDs, which are already widely discussed and known.

4. For the vibration feedback, how strong was it? The authors should provide the objective value of the vibration frequency, etc.

5. Is there any evidence that the authors set the type of corrective feedback as long vibration while short and intermittent vibration was used for the encouraging one? Furthermore, If these two are changed, would the current results be changed?

Author Response

This paper investigates the impact of corrective and encouraging vibrotactile feedback on monitoring and enhancing working posture. The research content of the paper is clear, and the research also has positive implications for the development of a posture feedback system. However, the research should be further explored based on the objectives of the study.

1. Why did the authors use corrective and encouraging vibrotactile feedback condition, not encouraging condition only? The reviewer concerns that there would be confounding effects of those two feedbacks, as it is described in the discussion.

Thanks for your comments! We discussed in the research team and the reason behind using both corrective and encouraging feedback was that we wanted the other group could receive enough amount of feedback from the system. Giving solely encouraging feedback might not ensure an adequate exposure to the feedback from the system. Since the study was of a pilot nature (we’ve emphasized it now in the manuscript), the choice of combining feedbacks was an explorative attempt. From our results we could see that it might be better to use solely encouraging feedback in future studies. At the same time, when designing future studies with solely corrective or encouraging feedback, it is worth considering how to design the work tasks so that both groups can get enough feedback from the system.

We have added explanation in the manuscript.

2. Although not significant, the results of comparing feedback performance must be explained from the start of the Results section, not the Appendix; as readers expect the experimental output based on the title and objective of the study. 

Thanks for the suggestion, we have now modified the text and moved the results of comparing groups in the results, shown by Table 2.

3. The first paragraph of the Introduction includes lengthy information regarding MSDs, while the core issue of this study is vibrotactile feedback. The authors might reduce (or remove) the first paragraph for MSDs, which are already widely discussed and known.

Thanks for the suggestion. Since it is a multi-disciplinary field, we included background information on MSDs and how training programs have been used in previous research to reduce MSDs in the original draft for readers who don’t know much about the field. Now we have reduced some information on the topic.

4. For the vibration feedback, how strong was it? The authors should provide the objective value of the vibration frequency, etc.

Thanks for the question. The long vibration for corrective feedback was designed to mimic a wasp and it lasted two seconds each time it is triggered. The intermittent vibration for encouraging feedback was designed to mimic bird song, and it included two repetitions of three short vibrations each lasing 0.1 seconds with 0.1 seconds pause, and a 0.5-second pause between repetitions.

Unfortunately, we cannot describe how strong the vibration was since the experienced vibration was affected by the placement of the whole vibration unit in the pockets, of which we lacked the information.

We have added the information available in the manuscript.

5. Is there any evidence that the authors set the type of corrective feedback as long vibration while short and intermittent vibration was used for the encouraging one? Furthermore, If these two are changed, would the current results be changed?

Thanks for the question. This choice was also of an explorative nature since there was a lack of knowledge in the research field. The team has discussed and tested the vibration choice and felt that the two types of feedback were easily distinguishable. The explanation given was a continuous vibration is mimicking a wasp, and the intermittent vibration is mimicking bird song.  

Regarding changing the vibrations used for the two feedback conditions, our results cannot answer the question. But maybe it won’t affect the results on performance as long as the participants could learn to differentiate / interpret the different feedback. Our choice of feedback aimed to facilitate their understanding / interpretation of corrective and encouraging feedback.

Reviewer 2 Report

The manuscript contains an interesting description of the results of research on the effects of 2 feedback training strategies on work technique, abut also individual motivation and user experience.

The title reflects the content of the manuscript. The keywords were chosen correctly. The executive summary contains content that allows you to get an idea of the topic and the results achieved.

The purpose of the research was correctly formulated. The research methods were adequately selected. The conditions in which they were carried out, their course and results were properly described. The methods of interpretation and presentation of the results were well chosen too. The material for publication was carefully prepared.

The division into chapters and its layout is correct, however there are repetitions in manuscript.

The described content is interesting and important however, the conducted research is rather of a pilot nature. It can be treated as an introduction to main research.

The authors should adequately describe the nature of the research (as preliminary, exploratory or pilot) and clearly define the need for further research steps.

The reason why it is difficult to consider it as a significant contribution to science with research value and authoritative character is the limitations related to the conditions of the research and the number of participants. Although they were described in detail in chapter 4.6. Limitations, however, this is not a sufficient argument. The research was carried out on a very small sample (10 persons), the problem is also the short period and only the laboratory environment as well as the lack of a control group.

Author Response

The manuscript contains an interesting description of the results of research on the effects of 2 feedback training strategies on work technique, abut also individual motivation and user experience.

The title reflects the content of the manuscript. The keywords were chosen correctly. The executive summary contains content that allows you to get an idea of the topic and the results achieved.

The purpose of the research was correctly formulated. The research methods were adequately selected. The conditions in which they were carried out, their course and results were properly described. The methods of interpretation and presentation of the results were well chosen too. The material for publication was carefully prepared.

The division into chapters and its layout is correct, however there are repetitions in manuscript.

Thanks for your comments!

The described content is interesting and important however, the conducted research is rather of a pilot nature. It can be treated as an introduction to main research.

Thank you for the suggestion. We agree that the study was of a pilot nature and have now added it in the manuscript.

The authors should adequately describe the nature of the research (as preliminary, exploratory or pilot) and clearly define the need for further research steps.

Thank you for the suggestion. We have added the explanation both in introduction and discussion.

The reason why it is difficult to consider it as a significant contribution to science with research value and authoritative character is the limitations related to the conditions of the research and the number of participants. Although they were described in detail in chapter 4.6. Limitations, however, this is not a sufficient argument. The research was carried out on a very small sample (10 persons), the problem is also the short period and only the laboratory environment as well as the lack of a control group.

Thank you for the comments. We consider this study as a pilot study towards more use of encouraging and accumulated feedback in work technique training, with both quantitative and qualitative evaluations. With the findings of our study, future research can include more participants and design the encouraging and/or accumulated vibrotactile feedback, first in a lab environment and then in field. Since there are multiple factors to modify in such studies, our results can provide some guidance in the design choices.

Reviewer 3 Report

1. Sec. 2.1. The study only used 10 participants for evaluation and eventually divided in to 2 groups. The sample size becomes the major problem of the generalization of the results.

2. Moreover, 9 females and 2 males were recruited. The gender effect seems not well controlled. The working posture is affecting by some critical factors, such as gender, stature…

3. Ln 126-131 The rest time set at 1 min. Is it enough? Does any learning effect occur with the setting of current experimental procedure? Maybe the re-test experiment should be conducted.

4. Ln 139 “The participants performed the tasks in a predetermined order.” Why predetermined order was used (not randomly order)?

5. Ln 186-191 the definition of each angle should be illustrated for better understanding.

6. Ln 226-228 The reasons for choosing the threshold of 30°、45° and 60° for arm needs a citation. The same query can be found for trunk angle.  

7. Ln 259 to 261. The statistics should be better to present in the context. (at least the p value)

8. Additionally, the writing style and grammar error should be sent to do the English editing. The format is not meet the academic writing basic level (e.g., the way of citations used in the context)

9. Overall, the quality of the current version is not recommended to publish. 

Author Response

  1. Thank you for the comments. We consider this study as a pilot study towards more use of encouraging and accumulated feedback in work technique training, with both quantitative and qualitative evaluations. With the findings of our study, future research can include more participants and design the encouraging and/or accumulated vibrotactile feedback, first in a lab environment and then in field. Since there are multiple factors (the encouraging/corrective nature of feedback, the time interval of providing feedback, the combination of concurrent and accumulated feedback, etc) to modify in such studies, our results can provide some guidance in the design choices. We have added some explanation in the manuscript.

  1. Thank you for the comments. We agree that it is a limitation in our study of a small sample size (8 females and 2 males) and we had drop out during the study due to participants having symptoms during Covid. While in order to see the effects of gender in the feedback training, a much larger group would be needed to ensure enough female and male in both groups. As an explorative pilot study focusing on encouraging and corrective accumulated feedback, our results can provide some guidance in the design choices for future studies with larger sample sizes.

  1. Since the tasks performed were quite simple, we consider a short one-minute break to be adequate for our purpose. The participants also reported no musculoskeletal discomforts after performing all the tasks. We have discussed in the “Study limitation” that a study design with a control group can be used to eliminate the learning effects. In a previous study, repeating a manual handling task without feedback for three work cycles showed no improvement in the work techniques. Therefore, considering the limited resources, the choice of two groups receiving the two types of accumulated vibrotactile feedback was prioritized in the study design. We agree that future studies should try to evaluate the long-term effects of using encouraging and/or accumulated vibrotactile feedback.

  1. Thanks for the question! The predetermined order of tasks was chosen since it is more common in industrial settings that manual handling tasks follow a specific order. In our study it also made the conditions more comparable to each other.

  1. Thanks for the suggestion. Now we have added Figure 5 in the manuscript to illustrate.

  1. Thanks for the suggestion. Now we have added relevant references in the manuscript.

  1. Thanks for the suggestion. Now we have added p values in the text, the Table B from Appendix was also moved into the results based on another reviewer's comment.

  1. Thanks for the comments. There might be some errors when importing references with different tools. We have used English language check tool and improved the ways to use citations in the context in the manuscript now.

9. Thanks for the comments. We have improved the manuscript according to all reviewers’ comments and are willing to further improve it if there is further comment. Hope the quality now is satisfying.

Round 2

Reviewer 1 Report

The author is satisfied.

Author Response

Thanks for your review!

Reviewer 3 Report

Thank you very much to the author's careful response for my previous concerns. However, these controls of experimental design still impacted directly to the result. At the same time, there is still a critical problem in gender differences and task settings (1 min) even if the author gives some statements as the reason. Unfortunately, the accuracy and application of the results under the restrictions cannot convince me. I still believe that it is necessary to increase experimental subjects and conducted a validation to give clear evidence.

Author Response

Thanks for your review! We will consider those espects in future studies following this pilot study.